# Synthesis of Novel Diketopyrrolopyrrole-Rhodamine Conjugates and Their Ability for Sensing Cu^2+^ and Li^+^

**DOI:** 10.3390/molecules27217219

**Published:** 2022-10-25

**Authors:** Carla Queirós, Vítor A. S. Almodôvar, Fábio Martins, Andreia Leite, Augusto C. Tomé, Ana M. G. Silva

**Affiliations:** 1LAQV-REQUIMTE, Department of Chemistry and Biochemistry, University of Porto, 4169-007 Porto, Portugal; 2LAQV-REQUIMTE, Department of Chemistry, University of Aveiro, 3810-193 Aveiro, Portugal

**Keywords:** rhodamine-based chemosensors, diketopyrrolopyrroles, copper complexes, lithium detection, displacement mechanism

## Abstract

The search for accurate and sensitive methods to detect chemical substances, namely cations and anions, is urgent and widely sought due to the enormous impact that some of these chemical species have on human health and on the environment. Here, we present a new platform for the efficient sensing of Cu^2+^ and Li^+^ cations. For this purpose, two novel photoactive diketopyrrolopyrrole-rhodamine conjugates were synthesized through the condensation of a diketopyrrolopyrrole dicarbaldehyde with rhodamine B hydrazide. The resulting chemosensors **1** and **2**, bearing one or two rhodamine hydrazide moieties, respectively, were characterized by ^1^H and ^13^C NMR and high-resolution mass spectrometry, and their photophysical and ion-responsive behaviours were investigated via absorption and fluorescence measurements. Chemosensors **1** and **2** displayed a rapid colorimetric response upon the addition of Cu^2+^, with a remarkable increase in the absorbance and fluorescence intensities. The addition of other metal ions caused no significant effects. Moreover, the resulting chemosensor-Cu^2+^ complexes revealed to be good probes for the sensing of Li^+^ with reversibility and low detection limits. The recognition ability of the new chemosensors was investigated by absorption and fluorescence titrations and competitive studies.

## 1. Introduction

In daily life, people are exposed to various contaminants and harmful substances that have a strong impact on human health by the inhalation of polluted air or by consuming contaminated water or food, among others. Analytical tools, such as optical sensors, that enable measurements through an optical colorimetric or fluorescent signal, can be an effective approach to detect and prevent the exposure of populations to these contaminated environments.

The use of diketopyrrolopyrrole (DPP) derivatives as fluorescent probes has been extensively explored in the last decade due to their high fluorescence quantum yields, exceptional photostability, and moderately high molar absorption coefficients [1,2,3]. The focus of these applications has been the tracing of biologically and environmentally important species, namely cations and anions [4,5,6,7]. The chemical versatility of the DPP skeletal unit allows for different functionalization to create a diverse range of DPP-based multifunctional dyes.

Rhodamines are also an important class of fluorescent molecules endowed with high molar absorptivity, intense fluorescence in the visible region, and an effective tuning of fluorescence emission through the spirolactam ring-opening mechanism, which is particularly relevant for designing acid- and metal-ion-sensing probes [8]. These singular properties make them very useful for many applications, namely as fluorescent markers in biotechnology, chemosensors for small molecule detection, and imaging in medicine [9,10,11]. Over the last few years, the design of chemosensors based on a displacement mechanism involving the interaction of an analyte with a specific coordination complex has emerged as a very attractive approach for the preparation of highly sensitive and selective ion chemosensors, with very promising results in real environment samples [12,13] and cell bioimaging [14]. Li’s group [15] provided the first rhodamine-based dye (I, Figure 1) for the indirect colorimetric detection of cyanide. Their approach involved the initial formation of the coloured complex I + Cu^2+^, as evidenced by the appearance of an absorption band at 555 nm. Upon the addition of cyanide, the colour faded to colourless as a result of the dissociation of the complex and the release of the initial ligand, achieving a detection limit of 0.013 ppm for cyanide. Other representative examples reported recently include: (a) a rhodamine 6G-phenylurea conjugate (II, Figure 1) for the recognition of acetate in the presence of Fe^3+^ [16]; (b) a squaraine-bis(rhodamine-B) chemosensor (III, Figure 1) for the recognition of cyanide upon the addition of Hg^2+^ [17]; (c) a rhodamine B-derived thiophene chemosensor (IV, Figure 1), which upon the addition of Cu^2+^ ions results in a strong absorption band at 554 nm, and whose complex (IV−Cu^2+^) was later used to detect cyanide [18]; and (d) a rhodamine-6G-based chemosensor characterized by a *turn-on* fluorescence response towards trivalent metal ions M^3+^ (M = Fe, Al and Cr) over mono- and divalent metal ions, which through the addition of cyanide provides its fluorescence quenching through the formation of the rhodamine spirolactam form (V, Figure 1) [19].

In this paper, we report the synthesis of two novel diketopyrrolopyrrole-rhodamine conjugates (**1** and **2**, Figure 1 and Figure 1) for the colorimetric and fluorescence sensing of Cu^2+^ and Li^+^. Two main objectives were drawn: (1) to prepare new photoactive conjugates through the combination of two classes of fluorophores (rhodamine and diketopyrrolopyrrole) and (2) to obtain new chemosensors with improved selectivity and sensitivity for ions compared to known probes.

## 2. Results and Discussion

### 2.1. Synthesis

Chemosensors **1** and **2** were obtained from the reaction of rhodamine B hydrazide (**RhoHyd**) with the DPP dicarbaldehyde (**DPP(CHO)_2_**) in refluxing ethanol (Figure 1). Both products were formed in this reaction. After being separated by preparative TLC, their structures were identified by NMR (Appendix A, Appendix A, Appendix A, Appendix A) and MS (Appendix A and Appendix A).

### 2.2. Absorption and Fluorescence Spectroscopy Measurements

The absorption and fluorescence properties of chemosensors **1** and **2**, and their precursors **RhoHyd** and **DPP(CHO)_2_**, were studied in three solvents: dichloromethane, ethanol, and acetonitrile. Figure 2 shows the UV-Vis and fluorescence spectra for **1**, **2**, **RhoHyd,** and **DPP(CHO)_2_** in anhydrous CH_2_Cl_2_. For **1**, **2,** and **DPP(CHO)_2_**, the spectra showed three absorbance bands at approximately 270, 320, and 493 nm (**2** also showed a “shoulder” at 345 nm) and a strong emission band at 573 nm, while **RhoHyd** exhibited only two absorbance bands at 275 and 316 nm and no significant emission, indicating that **RhoHyd** existed predominantly in the spirolactam form.

The main spectroscopic properties of chemosensors **1** and **2** in CH_2_Cl_2_, EtOH, and CH_3_CN are presented in Table 1, while the data for precursors are provided in Appendix A. Both the absorption and fluorescence properties of **1** and **2** presented major similarities with **DPP(CHO)_2_**, including large Stokes shifts and emission values above 500 nm, which was a good indication for their potential application as fluorescent probes [3]. The fluorescence quantum yield of **2** was slightly higher than that of **1** in CH_2_Cl_2_, but lower in EtOH and CH_3_CN. The values attained for both chemosensors in CH_2_Cl_2_ were about four times higher than those for the other solvents. This large difference was not observed for **DPP(CHO)_2_**, where the highest *Φ*_F_ was also observed for CH_2_Cl_2_ but only differed from EtOH and CH_3_CN by a factor of 1.04, meaning that the spectroscopic properties of chemosensors **1** and **2** were more sensitive to solvent polarity. This influence can also be observed in the lifetime values, which presented a decrease in value from CH_2_Cl_2_ to EtOH and CH_3_CN. Considering that chemosensors **1** and **2** exhibited larger Stokes shifts and fluorescence quantum yields in CH_2_Cl_2_, their sensing abilities were studied in this solvent. 

### 2.3. UV-Vis and Fluorescence: Solution Studies

The sensing abilities of chemosensors **1** and **2** were investigated by monitoring changes in the UV-Vis and fluorescence spectra upon the addition of various metal ions, including Li^+^, Na^+^, K^+^, Ag^+^, Ca^2+^, Ni^2+^, Cu^2+^, Zn^2+^, Pb^2+^, Al^3+^, Fe^3+^, and Ga^3+^, and anions, including F^−^, Cl^−^, Br^−^, I^−^, BF_4_^−^, ClO_4_^−^, HSO_4_^−^, NO_3_^−^, and PF_6_^−^. The initial experiments consisted of monitoring the colorimetric and fluorescence changes upon the addition of 5 equiv. of cations or anions to chemosensors **1** and **2**. Both chemosensors presented an orange coloration in CH_2_Cl_2_ and no changes in colour or spectra were observed after the addition of the tested anions (Appendix A). However, a strong colorimetric response was observed, both for **1** and **2**, in the presence of Cu^2+^—the solution became pink with an intense coloration (Figure 3). The fluorescence spectra of both chemosensors also changed similarly in the presence of this ion.

In the UV-Vis spectra of **1** and **2**, new absorption bands at 519 and 560 nm appeared in the presence of Cu^2+^, Al^3+^, and Ga^3+^, but with a much higher intensity for Cu^2+^. On the other hand, the absorption band at 493 nm practically disappeared in the presence of Fe^3+^. For **1**, under excitation at 493 nm, changes were observed in the emission band at 577 nm, with an increase for Cu^2+^, Al^3+^, and Ga^3+^ and a decrease (quenching) for Fe^3+^; less significant differences were observed for **2** (Appendix A). No colorimetric or spectroscopic changes were noticeable for Li^+^, Na^+^, K^+^, Ag^+^, Ca^2+^, Ni^2+^, Zn^2+^, and Pb^2+^. These results indicate that both chemosensors **1** and **2** can serve as naked-eye indicators for Cu^2+^.

For control purposes, the same experiments with metal ions (Na^+^, K^+^, Ca^2+^, Ni^2+^, Cu^2+^, Zn^2+^, Pb^2+^, Al^3+^, Fe^3+^, and Ga^3+^) and anions (F^−^, Cl^−^, Br^−^, I^−^, BF_4_^−^, ClO_4_^−^, HSO_4_^−^, NO_3_^−^, and PF_6_^−^) were performed for **RhoHyd** and **DPP(CHO)_2_**. None of these compounds showed changes in naked-eye colour or fluorescence (under UV light) in the presence of the selected anions (Appendix A).

However, it was observed that **DPP(CHO)_2_** lost its yellow coloration and fluorescence in the presence of Fe^3+^, with no alterations for the other metal ions, an indication that this compound could be a good chemosensor for Fe^3+^. On the other hand, **RhoHyd** acquired a pink coloration and high fluorescence (under UV light) in the presence of Cu^2+^, Al^3+^, and Fe^3+^, with a higher intensity for the first (Appendix A).

Considering these results and knowing that there is a continuous interest in finding new selective and sensitive chemosensors for Cu^2+^, we decided to deepen our study of the response of the conjugates towards Cu^2+^. Two different copper salts were tested—chloride and trifluoromethanesulfonate—as counter-ions, and the observed behaviour was similar. For chemosensor **1**, the addition of Cu^2+^ was performed up to 1 equiv. and for **2** up to 2 equiv. In the absorption spectrum of **1**, slight shifts for the bands at 274 nm (blueshift) and 323 nm (redshift) were observed and two new absorption bands appeared at 519 and 560 nm. The intensity of the new bands increased with increasing Cu^2+^ concentration (Figure 4, left). A similar behaviour was observed for chemosensor **2** (Appendix A). The presence of isosbestic points reflects the existence of only one intermediate complex [6,7]. Under excitation at 560 nm, a significant increase in the emission was observed for the band at 577 nm (Figure 4, right). For both absorption and emission spectra, the intensity values increased linearly with the Cu^2+^ concentration (Appendix A).

Apart from steady-state measurements, fluorescence lifetime data (Table 1) were also acquired in order to support the fluorescence enhancement presented here. Both chemosensors **1** and **2** presented a biexponential decay (*λ*_em_ = 577 nm) that combined a major long-lived decay and a minor short-lived decay. Chemosensor **1** showed fluorescence lifetimes of 0.73 ns and 3.25 ns, the latter being the most abundant with a 90.7% of relative abundance. Similarly, chemosensor **2** showed a slightly faster biexponential decay with lifetimes of 0.36 ns and 2.65 ns, with the long-lived lifetime having a contribution of 84.9%. In both cases, with increasing Cu^2+^ concentration, we observed a slight gradual increase in the long-lived decay values up to 3.37 ns and 2.91 ns for chemosensors **1** and **2**, respectively. This behaviour can be rationalized by the formation of a Cu^2+^ complex with **1** and **2** and the opening of the spirolactam ring of the rhodamine. This ring-opening mechanism leads to a slight increase in the chemosensors’ rigidity, thus favouring the radiative decay pathways over non-radiative decay and increasing the fluorescence intensity and lifetime of the chemosensor [20]. 

The sensitivity of the metal ion-induced signalling responses of the chemosensors, defined by the limits of detection (LOD) and limits of quantification (LOQ), was determined through the calibration curve method for the absorption and fluorescence spectral responses of **1** and **2** with Cu^2+^ at low concentrations. Equations (1) and (2) were employed for the determination of these values [21]:(1)LOD=3.3×Standard error×NSlope
(2)LOQ=10×Standard error×NSlope
where the *standard error* represents the standard deviation of the response, *Slope* is the slope of the calibration curve, *N* is the number of independent repetitions (*N* = 3), and the coefficients 3.3 and 10 are expansion factors obtained assuming a 95% confidence level [22]. The LOD values (and LOQ values, in parentheses) of Cu^2+^ detection for **1** and **2** were calculated to be 0.46 μmol dm^−3^ (1.40 μmol dm^−3^) and 0.16 μmol dm^−3^ (0.47 μmol dm^−3^), respectively, from the absorption spectral responses, and 0.33 μmol dm^−3^ (1.00 μmol dm^−3^) and 0.18 μmol dm^−3^ (0.55 μmol dm^−3^), respectively, from the fluorescence spectral responses. These limits are similar to others reported in the literature in the μmol dm^−3^ range; a comparison is presented in Table 2.

### 2.4. Competition Studies

An effective chemosensor requires a high selectivity for the target analyte over potentially competitive species. To assure this, competition experiments with potentially competitive metal ions were conducted using UV-Vis absorption and fluorescence spectroscopies. First, solutions of the complexes **1**+Cu^2+^ and **2**+Cu^2+^ were prepared by the addition of Cu^2+^ (1 or 2 molar equiv., respectively) to CH_2_Cl_2_ solutions of **1** and **2**. Then, other metal ions were added to the previous solution (1 equiv. for **1**+Cu^2+^ and 2 equiv. for **2**+Cu^2+^). A considerable absorbance decrease at 560 nm was observed upon the addition of Fe^3+^, Li^+^, Na^+^, and K^+^ to the solution of **1**+Cu^2+^ but no significant changes were observed for the other cations (Figure 5A). In terms of fluorescence emission, at 577 nm, a similar response occurred (quenching) for the mentioned cations, with fluorescence quenching values above 70%. Slightly lower quenching responses were also observed for Ag^+^, Al^3+^, and Ga^3+^ (Figure 5B). For the complex **2**+Cu^2+^, a decrease in absorption was detected for some monovalent cations, especially Li^+^, with a decrease in intensity above 80% (Figure 5C), while for fluorescence, the quenching effect was never above 40%. The sensitivity was more prominent for the trivalent and monovalent cations (Figure 5D).

Considering the numerous technological applications associated with Li^+^, the interest in developing new chemosensors for its detection is extremely high [28,29,30,31]. For that reason, we decided to carry out further titration studies using **1**+Cu^2+^ and **2**+Cu^2+^ complexes to detect Li^+^. The spectroscopic responses for **1**+Cu^2+^ are presented in Figure 6. The results clearly showed that the absorbance and fluorescence intensities were inversely proportional to the concentration of Li^+^. The same effect was observed for the complex **2**+Cu^2+^ (Appendix A).

To better understand the quenching behaviour of **1+Cu^2+^** and **2+Cu^2+^,** Stern-Volmer plots were applied according to Equation 3, [32] plotting *F_0/_F* as a function of the Li^+^ concentration (quencher, *Q*):(3)$$\frac{F_{0}}{F}=1+K_{SV}\;\left[Q\right]$$
where *F_0_* and *F* represent the fluorescence intensities in the absence or presence of the quencher (*Q*), respectively, and *K_SV_* is the Stern-Volmer constant.

The resulting plots showed a downward deviation from linearity towards the *x*-axis (Appendix A (A and C)). This behaviour is typical of systems where the quencher is only accessible to a fraction of the fluorophore. In the present study, this stabilization can be explained by the fact that in **1**+Cu^2+^ and **2**+Cu^2+^, there is a fraction of the fluorescence that corresponds to the emission of the diketopyrrolopyrrole unit, while the rhodamine-copper unit has its fluorescence quenched by interaction with Li^+^, thus leaving the fluorescence of diketopyrrolopyrrole unchanged. In order to linearize the relationship of the fluorescence intensity with the increasing concentration of the quencher, we used the modified Stern-Volmer equation, also called the Lehrer equation:(4)$$\frac{F_{0}}{\Delta F}=\frac{1}{f_{a}}+1/\left(f_{a}\;K_{SV}\;\left[Q\right]\right)$$

According to Equation 4, and by plotting *F_0_/*Δ*F* as a function of the Li^+^ concentration (quencher, *Q*) we obtained the Stern-Volmer constant of the accessible fraction (*K_SV_*) and the fraction of fluorescence that is accessible to the quencher (*f_a_*) (Table 3 and Appendix A (B and D)).

To better understand the nature of the interaction under study, we resorted to fluorescence lifetime data. As before, both **1**+Cu^2+^ and **2**+Cu^2+^ showed a biexponential decay (*λ*_em_ = 577 nm) that combined a major long-lived decay and a minor short-lived decay. In both cases, with increasing Li^+^ concentration we observed a small and gradual decrease in the more abundant long-lived decay to values closer to those found prior to the addition of copper. For **1**+Cu^2+^, a decrease in the long-lived lifetime occurred from 3.37 ns to 3.27 ns and for **2**+Cu^2+^ from 2.91 ns to 2.76 ns (Figure 7). Due to the decrease in the fluorescence lifetimes along with the quenching of fluorescence intensity, we hypothesised that the interaction of Li^+^ with the **1**+Cu^2+^ and **2**+Cu^2+^ complexes is of a dynamic nature. Therefore, the quenching constant presented in Table 3 is actually *K_D_* and not *K_SV_*. The fluorescence decrease originates from collisions of the quencher with the fluorophore in the excited state, thereby displacing the copper and returning the chemosensor to its original form (**1** and **2**), leading to a preference for the non-radiative decay pathways over radiative decay. These quenching results (Table 3) point to the fact that **1**+Cu^2+^ presents a higher dynamic quenching constant (*K_D_*) compared to **2**+Cu^2+^; therefore, **1**+Cu^2+^ not only presents the highest fractions of fluorescence intensity available to the quencher but also is more efficiently quenched by Li^+^.

Different lithium salts, namely Li(CH_3_COO) and LiBr, were used to assess the counter-ion influence on the behaviour of chemosensors here reported. The obtained spectra (Appendix A and Appendix A) showed in both cases an evident decrease in absorption and a corresponding alteration in colour (not shown) and fluorescence with successive additions of the Li salts, which supported the notion that the observed behaviour was due to Li cations and not a counter-ion effect.

The behaviour of the Cu^2+^ complexes in the presence of cyanide anions was also studied but, in general, the response obtained was not linearly correlated with the CN^−^ concentration (Appendix A and Appendix A). For **1**+Cu^2+^, a *K_SV_* value of approximately 538 dm^3^ mol^−^^1^ (Appendix A) was determined and, considering the low value, no further studies were performed. Nevertheless, the CN^−^ concentration range detectable by **1**+Cu^2+^ was below the toxicity limit for cyanide (19 μmol dm^−3^). Looking at these results, it can be stated that both **1**+Cu^2+^ and **2**+Cu^2+^ have higher sensitivity to Li^+^ than to CN^−^_,_ with **1**+Cu^2+^ being more sensitive than **2**+Cu^2+^.

### 2.5. Mechanism Studies

According to the presented results, we propose that the interaction of chemosensors **1** and **2** with Cu^2+^ leads to the formation of the corresponding 1:1 and 1:2 complexes; this was verified via a colour change and a change in the fluorescence spectra. The resulting complexes can be easily reversed by the displacement strategy, thereby recovering the pristine chemosensor. To support this hypothesis, a solution of **2** in dichloromethane was treated with 2 equiv. of Cu^2+^, which resulted in an immediate colour change from orange to pink. After a stabilization period (15−30 min), the solution was washed with water and the colour returned to the original orange. The organic phase was separated and dried and a ^1^H NMR analysis was carried out, which showed the structure of the original chemosensor **2**. This result confirms that the interaction of the ligand with Cu^2+^ is weak and that a simple wash with water is sufficient to rapidly dissociate the complex and recover the initial ligand. Further studies were carried out using ^1^H NMR and EPR. Upon the addition of 2 equiv. of Cu^2+^ to a solution of **2** in CDCl_3_, the ^1^H NMR spectrum revealed the general broadening of all signals due to the paramagnetic nature of the **2**+Cu^2+^ complex (Figure 8A). By EPR (Figure 8C), it was possible to confirm the formation of the **2**+Cu^2+^ complex through the appearance of the EPR signal characteristic of Cu^2+^ complexes [33]. The subsequent addition of Li^+^ caused the dissociation of the complex, as evidenced by the disappearance of the EPR signal and the recovery of a well-resolved ^1^H NMR spectrum (Figure 8B) identical to that of chemosensor **2** alone.

These results suggest that when another targeted analyte, especially Li^+^, is added to the chemosensor-Cu^2+^ complexes, a receptor-analyte displacement reaction takes place, resulting in a spectroscopic response [34,35] and the reversibility behaviour of the chemosensors. We anticipate that the incorporation of these chemosensors in solid matrixes could open a broader prospect for the chemosensors’ application, even in aqueous systems, for detecting environmentally and biologically important species. To demonstrate this hypothesis, we prepared three test strips by immersing filter paper in a dichloromethane solution of chemosensor **2** (2.0 × 10^−6^ mol dm^−3^) and drying them in air. As shown in Figure 9, when the test strip was immersed in an aqueous solution of Cu^2+^ (6.0 × 10^−3^ mol dm^−3^) and dried for 10 min in an oven at 80 °C, a significant colour change was observed from colourless to magenta (more pronounced under UV lamp); after immersion in an aqueous solution of Li^+^ (6.0 × 10^−3^ mol dm^−3^), the magenta colour faded to pale, which demonstrated a possible practical application of the chemosensor **2** for detecting Cu^2+^ and Li^+^ in aqueous solutions. 

## 3. Materials and Methods

Rhodamine B, hydrazine hydrate, tetrabutylammonium salts (nitrate, perchlorate, tetrafluoroborate, hexafluorophosphate, hydrogen sulphate, fluoride, chloride, bromide, and iodide), sodium cyanide, iron(III) trifluoromethanesulfonate, copper(II) chloride, copper(II) trifluoromethanesulfonate, nitrate hydrate salts (calcium(II), aluminium(III), lead(II), gallium(III), and silver(I)), acetate hydrate salts (zinc(II), nickel(III), potassium, and sodium), lithium hydroxide, lithium acetate and lithium bromide, anhydrous dichloromethane, anhydrous methanol, anhydrous acetonitrile, and absolute ethanol were purchased from Merck Life Science (Algés, Portugal) and were used as received.

Analytical thin-layer chromatography (TLC) was carried out on sheets precoated with silica gel (Merck 60, 0.2 mm thick). Preparative thin-layer chromatography was carried out on 20 cm × 20 cm glass plates precoated with a layer of silica gel 60 (0.5 mm thick) and dried in an oven at 100 °C for 12 h. ^1^H and ^13^C NMR spectra were recorded using a Bruker Avance 300 or Bruker Avance 500 both from Strasbourg (France). CDCl_3_ was used as a solvent and tetramethylsilane (TMS) as an internal reference. The chemical shifts (*δ*) were expressed in ppm and the coupling constants (*J*) in hertz (Hz). High-resolution MS analysis was carried out using electrospray ionization (ESI) in an LTQ Orbitrap XL instrument (Thermo Scientific, Waltham, MA, USA) with the following ESI source parameters: an electrospray needle voltage of 3 kV, sheath gas nitrogen 5, capillary temperature of 275 °C, capillary voltage of 37 V, and tube lens voltage of 120 V. Ionization polarity was adjusted according to the sample.

The electronic absorption spectra were recorded with a Shimadzu UV 3600 UV-Vis-NIR equipped with a Shimadzu TCC controller (Santa Clara, CA, USA) using quartz cells with a 1 cm path length. Steady-state fluorescence measurements were carried out in a Varian spectrofluorometer, Cary Eclipse model (Agilent Technologies, Santa Clara, CA, USA), equipped with a constant-temperature cell holder (Peltier single-cell holder) with a 5 nm slit width for excitation and emission. Time-resolved fluorescence data were acquired with a DeltaFlex time-correlated single-photon counting (TCSPC) system attached to a FluoroHub A+ controller, both from Horiba Jobin Yvon (Kyoto, Japan). Excitation was provided by a Horiba NanoLED source at 562 nm and the emission was recorded at the maximum wavelength of the samples. The lamp profile was recorded by measuring a scattering solution (dilute solution of LUDOX in water) and the results were taken into account when performing the data analysis using a nonlinear least squares iterative convolution method with the DAS6 v6.5 decay analysis software provided by Horiba Jobin Yvon. All photophysical assays were performed under controlled temperature conditions (25 °C) using the maximum *λ*_abs_ and the appropriate *λ* range for each chemosensor and considering the different solvents used. Chemosensor solutions were prepared in anhydrous CH_2_Cl_2_ (a new solution was prepared for each study or assay). Fluorescence quantum yields were determined using a Quantaurus-QY C11347-11 spectrometer, absolute luminescence quantum yield spectrometer (Hamamatsu Photonics, Massy, France).

EPR spectra were recorded using an X-band (9 GHz) Bruker ELEXSYS E 500 spectrometer (Bruker Biospin Group, Ettlingen, Germany) equipped with a variable temperature unit (ER 4B1 VT), available at Laboratório de Ressonância Paramagnética Electrónica (Centro de Materiais da Universidade do Porto), and obtained under the following general experimental conditions: microwave frequency of 100 kHz, microwave power of 20 mW, modulation amplitude of 8 G, gain of 60 dB, and acquisition time of 300 s at RT. The samples were prepared by the dissolution of the compound in chloroform in a capillary that was placed in a quartz tube.

### 3.1. Synthesis

Rhodamine B hydrazide (**RhoHyd**) and 4’,4’’’-(3,6-dioxo-2,5-dipentyl-2,3,5,6-tetrahydropyrrolo[3,4-*c*]pyrrole-1,4-diyl)bis([1,10-biphenyl]-4-carbaldehyde) (**DPP(CHO)_2_**), were synthesized according to methods reported in the literature [36,37].

### 3.2. Synthesis of Chemosensors **1** and **2**

A mixture of **RhoHyd** (70 mg, 0.154 mmol) and **DPP(CHO)_2_** (50 mg, 0.077 mmol) in ethanol was heated at reflux for 24 h. The solvent was evaporated under reduced pressure, the resulting solid was dissolved in dichloromethane, and the two products were purified via preparative TLC using dichloromethane as an eluent. Chemosensors **1** and **2** were dried and obtained in 33% (27 mg) and 36% (42 mg) yields, respectively.

Chemosensor **1**: ^1^H NMR (300 MHz, CDCl_3_) *δ* (ppm): 10.08 (s, 1H), 8.70 (s, 1H), 7.89–8.03 (m, 7H), 7.79–7.83 (m, 4H), 7.65–7.74 (m, 4H) 7.56–7.59 (m, 2H), 7.45–7.52 (m, 2H), 7.12–7.15 (m, 1H), 6.55 (d, *J* = 9 Hz, 2H), 6.46–6.47 (d, *J* = 3 Hz, 2H), 6.25 (dd, *J* = 9, 3 Hz, 2H), 3.72–3.80 (m, 4H), 3.30–3.37 (q, *J* = 6.0 Hz, 8H) 1.63–1.68 (m, 4H), 1.21–1.34 (m, 8H), 1.17 (t, *J* = 7.5 Hz, 12 H), 0.86–0.92 (m, 6H). ^13^C NMR (100 MHz, CDCl_3_) *δ* (ppm): 12.6, 13.9, 22.2, 28.9, 29.7, 42.1, 44.4, 66.1, 97.9, 106.0, 108.0, 109.9, 110.3, 123.4, 123.9, 127.0, 127.1, 127.4, 127.78, 127.83, 128.07, 128.13, 128.27, 128.34, 129.3, 129.4, 130.4, 133.5, 135.3, 135.7, 140.9, 142.2, 143.3, 145.9, 146.3, 147.5, 148.5, 149.0, 151.8, 153.2, 162.7, 162.9, 165.1, 191.8. HRMS (ESI) *m/z*: calcd. For C_70_H_71_N_6_O_5_^+^ [M + H]^+^ 1075.5480, found 1075.5465.

Chemosensor **2**: ^1^H NMR (300 MHz, CDCl_3_) *δ* (ppm): 8.70 (s, 2H), 7.99–8.02 (m, 2H), 7.88–7.92 (m, 4H), 7.66–7.75 (m, 8H) 7.56–7.59 (m, 4H), 7.47–7.52 (m, 4H), 7.12–7.15 (m, 2H), 6.55 (d, *J* = 9 Hz, 4H), 6.47 (d, *J* = 3 Hz, 4H), 6.25 (dd, *J* = 9, 3 Hz, 4H), 3.72–3.80 (m, 4H), 3.33 (q, *J* = 6.0 Hz, 16H) 1.63–1.68 (m, 4H), 1.21–1.34 (m, 8H), 1.16 (t, *J* = 7.5 Hz, 24 H), 0.86–0.92 (m, 6H). ^13^C NMR (100 MHz, CDCl_3_) *δ* (ppm): 13.5, 14.8, 23.1, 29.8, 30.6, 43.0, 45.2, 66.9, 98.8, 106.9, 108.9, 110.9, 124.3, 124.8, 127.9, 128.1, 128.2, 128.9, 129.0, 129.2, 130.1, 130.2, 134.3, 136.2, 141.8, 144.1, 147.3, 148.9, 149.9, 152.7, 154.1, 163.7, 165.9. HRMS (ESI) *m/z*: calcd. For C_98_H_101_N_10_O_6_^+^ [M + H]^+^ 1514.7934, found 1514.7927.

### 3.3. UV-Vis and Fluorescence Spectra Measurements

The stock solutions of analytes (cations and anions) and dyes (**RhoHyd**, **DPP(CHO)_2_,** and chemosensors **1** and **2**) were prepared in anhydrous methanol and dichloromethane, respectively. The compounds were characterized in different solvents, including anhydrous dichloromethane, anhydrous acetonitrile, and absolute ethanol. The solutions of analytes and dyes were then diluted appropriately to the working concentrations. In preliminary competitive experiments, the solutions of chemosensors **1** and **2** (2 mL, CH_2_Cl_2_, 10^−5^ mol dm^−3^) were added to a quartz cuvette with a 1 cm optical path length; then, aliquots of the cation and anion solutions (5 equiv. CH_2_Cl_2_) were added into the quartz cell using a micropipette. The same procedure was used for the titration experiments, but in this case up to 1 equiv. was used for chemosensor **1** and 2 equiv. for **2**. The added volume of cation or anion diluted solutions was less than 100 µL with the purpose of keeping the total volume of testing solution constant without significant change. Solutions of various concentrations containing the chemosensors and increasing concentrations of cations or anions were prepared separately. The UV-Vis and fluorescence spectra of these solutions were recorded 5 minutes after analyte addition. For fluorescence measurements, three excitation wavelengths were used (493, 519, and 560 nm), and the emission range was chosen as a function of the excitation value up to 800 nm (both the excitation and emission slit widths were kept at 5 nm).

## 4. Conclusions

In summary, we developed two novel chemosensors (**1** and **2**) which resulted from the condensation of a **DPP(CHO)_2_** derivative with one or two rhodamine hydrazide units. Both chemosensors exhibited a colorimetric and fluorescence response for the recognition of Cu^2+^, with a colour change from orange to pink and a prominent increase of 15.4 times (1 equiv. Cu^2+^) and 4.6 times (2 equiv. Cu^2+^) in emission for **1** and **2**, respectively. LOD values of 0.46 µmol dm^−3^ and 0.16 µmol dm^−3^ (fluorescence response) were determined for **1** and **2**, respectively. The excellent selectivity and sensitivity of the chemosensors to Cu^2+^ makes them promising probes for Cu^2+^ detection. When **1**+Cu^2+^ and **2**+Cu^2+^ complexes were used in the presence of CN^−^, the sensitivity was low; nevertheless, **1**+Cu^2+^ presented some quenching in absorption and emission for low concentrations of CN^−^. On the other hand, when **1**+Cu^2+^ and **2**+Cu^2+^ complexes were used in the presence of Li^+^, colorimetric and fluorescence responses were observed, particularly for **1**+Cu^2+^. These responses corresponded to a change in colour (from pink to orange) and a decrease in the absorbance and fluorescence emissions. A dynamic quenching behaviour was observed for both complexes in the presence of Li^+^, with *K_D_* values of 18.1 × 10^5^ and 8.00 × 10^5^ dm^3^mol^−1^ for **1**+Cu^2+^ and **2**+Cu^2+^, respectively. Therefore, the spectroscopic and colorimetric responses of the Cu^2+^ complexes towards Li^+^ make these complexes suitable for the detection of this cation.

## Data Availability

The data presented in this study are contained within the article and are also available in the Appendix A.

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
