# Peer review of "Synthesis of Novel Diketopyrrolopyrrole-Rhodamine Conjugates and Their Ability for Sensing Cu2+ and Li+"

_molecules, 2022, doi:10.3390/molecules27217219_

Round 1

Reviewer 1 Report

The search for accurate and sensitive methods to detect ions' effects on human health and on the environment are so popular.

In this study, the authors present a new platform for the efficient sensing of Cu2+ and Li+ cations. 

In this respect, this study is valuable and can be accepted as a result of some small suggestions below.

English should be checked thoroughly and some typos should be corrected (for exp. LOQ, LOD, etc.).

NMR spectra are not preferred because of the paramagnetic properties of Cu ions. EPR is sufficient to demonstrate binding, but FT-IR can be recommended for the ligand-ion interaction mechanism.

I have found it quite successful as giving all the formulas in the text provides a practical gain for researchers who want to acquire basic knowledge in this field. However, although it is not very necessary, I think it would be better if this detailed information and formulas could be given in the experimental part.

Researchers have used CuCl2 salts. On the other hand, the Cu2+ forms that can be detected are not only in the form of chloride salt.

Therefore, with different copper salts (CuSO4, Cu(NO3)2, etc.) which have become an important parameter of effective sensor work, results should be scanned only on UV-Vis and/or PL.

Author Response

Response to Reviewer # 1

The search for accurate and sensitive methods to detect ions' effects on human health and on the environment are so popular. In this study, the authors present a new platform for the efficient sensing of Cu2+ and Li+ cations. In this respect, this study is valuable and can be accepted as a result of some small suggestions below.

  • English should be checked thoroughly and some typos should be corrected (for exp. LOQ, LOD, etc.).

Response: The correction of typos has been done throughout the text.

  • NMR spectra are not preferred because of the paramagnetic properties of Cu ions. EPR is sufficient to demonstrate binding, but FT-IR can be recommended for the ligand-ion interaction mechanism.

Response: We would like to thank the reviewer's comment and suggestion. In fact, we recognize that the NMR spectra of the Cu(II) complex are affected by the paramagnetic nature of the metal ion, but we believe that it can be a good tool to prove that the addition of Li(I) led to the recovery of the integral structure of the ligand. Considering that the studies were carried out only in solution, the use of FT-IR is impaired in this case.

  • I have found it quite successful as giving all the formulas in the text provides a practical gain for researchers who want to acquire basic knowledge in this field. However, although it is not very necessary, I think it would be better if this detailed information and formulas could be given in the experimental part.

Response: The authors are grateful for the reviewer’s comment, but we believe that the presence of the formulas facilitates the reading and understanding of the manuscript and therefore decided to maintain them in the text.

  • Researchers have used CuCl2 On the other hand, the Cu2+ forms that can be detected are not only in the form of chloride salt. Therefore, with different copper salts (CuSO4, Cu(NO3)2, etc.) which have become an important parameter of effective sensor work, results should be scanned only on UV-Vis and/or PL.

Response: We recognize the importance of using different copper salts to determine the sensor effectiveness. In fact, one of the first trials performed (but not reported) involved a series of trials with another copper salt, copper(II) trifluoromethanesulfonate, without identifying significant changes in the interaction of ligands with the metal ion. To provide some additional information to the reader, the following short description was included in the manuscript text (lines 154-156):

“Two different copper salts were tested – chloride and trifluoromethanesulfonate as counter-ions – and the behaviour was similar between them.”

Likewise, the interference of anions was evaluated and the results showed that none interfered significantly in the absorption and/or emission of the chemosensors as stated in the text. The list of anions tested included: nitrate, perchlorate, tetrafluoroborate, hexafluorophosphate, hydrogen sulfate, fluoride, chloride, bromide and iodide.

Reviewer 2 Report

In the work "Synthesis of novel diketopyrrolopyrrole–rhodamine conjugates 2 and their ability for sensing Cu2+ and Li+" the authors propose two new sensors on Cu2+ and explore the performance of these sensors in the presence of a wide set of other ions. The manuscript is very well written, has clear structure and contains only minor amount of typos. In my view, this work could be considered for publication in Molecules provided the following issues are resolved.

1. The authors report "the spectroscopic and colorimetric responses of the Cu2+ complexes towards Li+" and suggest application of these complexes for detection of Li+. However, according to Materials and Methods section, LiOH was used as a source of Li+ ion in this work, while NaOOCH3 and KOOCH3 were used as a source of Na+ and K+ ions respectively. There were no other hydroxides among compounds used for the spectroscopic studies, according to Materials and Methods section. Can the observed responses be due to the presence of -OH anions rather than Li+ cations? This is my major concern regarding the presented work.

2. Please revise the data on the NMR spectra of compounds 1 and 2 (lines 362-378). The intensity of the signals in 1H NMR spectrum of 1 exceeds the expected number of hydrogen atoms. The number of signals in 13C NMR spectrum of 2 exceeds the expected value. The number of signals in 13C NMR spectrum of 1 is insufficient for the proposed structure. Please comment on that.

3. Figure 8: what is the source of proton in the depicted structure for 2+Cu2+ complex? What is the driving force for Li+ ion to release the ligand from 2+Cu2+ complex?

4. Please report the wavelength used to normalize the absorption spectra.

5. Figure 5, panels C and D: please check the correctness of the annotation of the second bars (Cu).

6. Please describe all the parameters used in the equations.

7. A few typos found:

Figure 1, legend: "on"->"in"

Lines 273-274: please revise the phrase "being verified that..."

Line 326: "with on" -> "with"

Author Response

Response to Reviewer # 2

In the work "Synthesis of novel diketopyrrolopyrrole–rhodamine conjugates and their ability for sensing Cu2+ and Li+" the authors propose two new sensors on Cu2+ and explore the performance of these sensors in the presence of a wide set of other ions. The manuscript is very well written, has clear structure and contains only minor amount of typos. In my view, this work could be considered for publication in Molecules provided the following issues are resolved.

  • The authors report "the spectroscopic and colorimetric responses of the Cu2+ complexes towards Li+" and suggest application of these complexes for detection of Li+. However, according to Materials and Methods section, LiOH was used as a source of Li+ ion in this work, while NaOOCH3 and KOOCH3 were used as a source of Na+ and K+ ions respectively. There were no other hydroxides among compounds used for the spectroscopic studies, according to Materials and Methods section. Can the observed responses be due to the presence of -OH anions rather than Li+ cations? This is my major concern regarding the presented work.

Answer: Thank you for the comment and alert of the reviewer concerning the use of LiOH salt in the studies performed. In fact, we selected the LiOH salt due to its high solubility in the solvent (methanol). Nevertheless, we understand the reviewer’s concerns and addressed the issue by performing two additional studies using:

  1. the Li(CH3COO) salt, considering that the acetate was the anion used in the studies for Na+ and K+ cations and
  2. the LiBr salt to discard the alkaline nature associated to OH- and CH3COO- ions.

These additional studies were performed with chemosensor 2 considering the low amount of the chemosensor 1 available in the laboratory at the present moment. The results obtained are depicted in Figures S14 and S15 (which were added in the Supporting Information) for Li(CH3COO) and LiBr, respectively, and show for both metal salts a similar behaviour to that observed with the hydroxide salt, with a quenching in color and fluorescence with successive additions of the Li salts. These results support that the behaviour observed is due to Li+ and not a counter-ion effect. To further acknowledge this issue, the following paragraph was added to the manuscript:

“Different lithium salts, namely Li(CH3COO) and LiBr, were used to assess the counter-ion influence on the behavior of chemosensors. The obtained spectra (Figures S14 and S15) showed in both cases an evident decrease of absorption – and the corresponding alteration in colour (not shown) – and fluorescence with successive additions of the Li salts, which supports that the observed behavior is due to Li cation and not a counter-ion effect.”

  • Please revise the data on the NMR spectra of compounds 1 and 2 (lines 362-378). The intensity of the signals in 1H NMR spectrum of 1 exceeds the expected number of hydrogen atoms. The number of signals in 13C NMR spectrum of 2 exceeds the expected value. The number of signals in 13C NMR spectrum of 1 is insufficient for the proposed structure. Please comment on that.

Answer: Thank you for the alert. As requested, all NMR spectra were corrected (corresponding to Figures S1, S2 and S4, in Supporting Information). It is important to note that compound 1 is asymmetrical, therefore, it will have more signals in the 13C NMR spectrum compared to compound 2.

  • Figure 8: what is the source of proton in the depicted structure for 2+Cu2+ complex?

Answer: We would like to thank the alert given by the reviewer. Indeed, we consider the protonated structure of the 2+Cu2+ complex depicted in Figure 8, but we acknowledge that the reviewer is right because there is no evidence of the protonation of the complex. In this mode, we proceed to the correction of the structure of Figure 8.

  • What is the driving force for Li+ ion to release the ligand from 2+Cu2+ complex?

Answer: Although we have not carried out studies that support the driving force for the release of Li+, our proposal is that the dissociation of the complex should occur, resulting in the recovery of the initial ligand in the spirocyclic form, as indicated by the NMR and EPR analyses.

  • Please report the wavelength used to normalize the absorption spectra.

Answer: The wavelength used to normalize the absorption spectra was the maximum wavelength for each compound (322 nm for DPP(CHO)2; 275 nm for RhoHyd; 322 nm for 1; and 273 nm for 2). The same reasoning was applied for the emission. To clarify this issue, the following sentence was added to the caption.

“The normalizations were done using the maximum wavelength for each compound”

  • Figure 5, panels C and D: please check the correctness of the annotation of the second bars (Cu).

Answer: The annotations were checked and corrected as presented in the Figure 5 (for chemosensor 1 equiv. of Cu(II) was added, while 2 equiv. were added to chemosensor 2).

  • Please describe all the parameters used in the equations.

Answer: The missing parameters were added to the manuscript text. New sentences were added to the text to complement the missing information:

  • “where the standard error represents the standard deviation of the response, Slope is the slope of the calibration curve, N is the number of independent repetitions (N=3) and the coefficients 3.3 and 10 are expansion factors obtained assuming a 95% confidence level”;
  • “where F0 and F represent the fluorescence intensities in the absence and presence of the quencher (Q), respectively, and KSV is the Stern-Volmer constant.”

The corresponding references were also added.

  • A few typos found: Figure 1, legend: "on"->"in", Lines 273-274: please revise the phrase "being verified that...", Line 326: "with on" -> "with"

Answer: The typos mentioned have been corrected.

Reviewer 3 Report

Dear Editor,

I have read the manuscript entitled: “Synthesis of novel diketopyrrolopyrrole–rhodamine conjugates 2 and their ability for sensing Cu2+ and Li+ ” there are several weaknesses that make publication possible only after minor revision.

Line 102: The authors say "Stokes shifts and emission values above 500 nm,". Please check this value as it is different from the one in Table 1.

Line 128: In Fig.4, the absorption band is represented as 519 nm, while in the text (Line 128) it is 520 nm.

Could the authors explain why in Figure 3 they also inserted Photographs of the solutions under a 365 nm lamp.

Could the authors introduce the corresponding references for equations (1)-(3) used in the text.

Minor points: Line 16: were synthesised = were synthesized; Figure 5: Bar plot = Bar plots

Author Response

Response to Reviewer # 3

I have read the manuscript entitled: “Synthesis of novel diketopyrrolopyrrole–rhodamine conjugates and their ability for sensing Cu2+ and Li+ there are several weaknesses that make publication possible only after minor revision.

  • Line 102: The authors say "Stokes shifts and emission values above 500 nm,". Please check this value as it is different from the one in Table 1.

Response: a comma was added in order to clarify the sentence “… including large Stokes shifts, and emission values above 500 nm”.

  • Line 128: In Fig.4, the absorption band is represented as 519 nm, while in the text (Line 128) it is 520 nm.

Response: The correction to 519 nm was done in the text.

  • Could the authors explain why in Figure 3 they also inserted Photographs of the solutions under a 365 nm lamp.

Response: The photographs of the solutions under the 365 nm lamp are presented to illustrate the changes in the fluorescence emission in the absence and presence of the analyte, similar results are also observed if a 254 nm lamp is used. In order to be coherent with figure 8, the caption was changed to: Photographs of the solutions under daylight (top) and under a UV lamp (bottom), for 1 (A) and 2 (B) in the presence of 5 equiv. of cations (Al3+, Fe3+, Ga3+, Cu2+, Ca2+, Zn2+, Ni2+, Pb2+, Na+, K+, Li+ and Ag+) in CH2Cl2, 25 °C.

  • Could the authors introduce the corresponding references for equations (1)-(3) used in the text.

Answer: The references for the equations were added to the text.

  • Minor points: Line 16: were synthesised = were synthesized; Figure 5: Bar plot = Bar plots

Answer: The corrections were done accordingly.

Round 2

Reviewer 2 Report

The authors have addressed my concerns, and I am glad to recommend this manuscript for publication in Molecules. As a minor remark: please check the description of the NMR spectra in p. 13. My guess is that "7.89-8.03 (m, 8H)" (line 380) should be "7.89-8.03 (m, 7H)", according to S1.

Author Response

The authors have addressed my concerns, and I am glad to recommend this manuscript for publication in Molecules. As a minor remark: please check the description of the NMR spectra in p. 13. My guess is that "7.89-8.03 (m, 8H)" (line 380) should be "7.89-8.03 (m, 7H)", according to S1

Answer: Thank you for the alert. The correction was made accordingly